# Rapid SABRE Catalyst Scavenging Using Functionalized Silicas

**DOI:** 10.3390/molecules27020332

**Published:** 2022-01-06

**Authors:** Thomas B. R. Robertson, Leon J. Clarke, Ryan E. Mewis

**Affiliations:** 1Department of Natural Sciences, Faculty of Science and Engineering, Manchester Metropolitan University, Chester Street, Manchester M1 5GD, UK; l.clarke@mmu.ac.uk (L.J.C.); r.mewis@mmu.ac.uk (R.E.M.); 2School of Chemistry, University of Southampton, Southampton SO17 1BJ, UK

**Keywords:** hyperpolarisation, NMR, SABRE, catalyst deactivation, solid-supported silicas

## Abstract

In recent years the NMR hyperpolarisation method signal amplification by reversible exchange (SABRE) has been applied to multiple substrates of potential interest for in vivo investigation. Unfortunately, SABRE commonly requires an iridium-containing catalyst that is unsuitable for biomedical applications. This report utilizes inductively coupled plasma-optical emission spectroscopy (ICP-OES) to investigate the potential use of metal scavengers to remove the iridium catalytic species from the solution. The most sensitive iridium emission line at 224.268 nm was used in the analysis. We report the effects of varying functionality, chain length, and scavenger support identity on iridium scavenging efficiency. The impact of varying the quantity of scavenger utilized is reported for the three scavengers with the highest iridium removed from initial investigations: 3-aminopropyl (**S_1_**), 3-(imidazole-1-yl)propyl (**S_4_**), and 2-(2-pyridyl) (**S_5_**) functionalized silica gels. Exposure of an activated SABRE sample (1.6 mg mL^−1^ of iridium catalyst) to 10 mg of the most promising scavenger (**S_5_**) resulted in <1 ppm of iridium being detectable by ICP-OES after 2 min of exposure. We propose that combining the approach described herein with other recently reported approaches, such as catalyst separated-SABRE (CASH-SABRE), would enable the rapid preparation of a biocompatible SABRE hyperpolarized bolus.

## 1. Introduction

Hyperpolarisation techniques are regularly employed to overcome the inherent sensitivity issue that is associated with NMR, and by extension, MRI. The inherent sensitivity arises due to the small population differences of the energy states that it probes. Dynamic nuclear polarization (DNP) [1,2,3], quantum-rotor induced polarisation [4], spin-exchange optical pumping (SEOP) [3,5], and *para*hydrogen induced polarization (PHIP) [2,3], are hyperpolarisation methods that lead to improved population differences. Thus, these methods are extensively used for the analysis and detection of metabolites or pharmaceuticals [6,7], catalytic intermediates [8,9,10], and for medical imaging purposes [11,12,13,14,15].

PHIP utilizes *para*hydrogen, a nuclear singlet and spin isomer of hydrogen, as a source of polarisation. In PHIP, *para*hydrogen is integrated into the substrate of interest through a hydrogenation reaction leading to a substrate that is chemically changed. The result of polarisation transfer creates a non-Boltzmann distribution of nuclear spins in the analyte, thus meaning the signals are noticeably enhanced in the ^1^H-NMR spectrum. Therefore, precursors possessing the correct functionality need to be prepared in order for the polarised molecule to be produced. A now established non-hydrogenative *para*hydrogen-based technique is Signal Amplification By Reversible Exchange (SABRE) [16]. Since this technique is non-hydrogenative, the analyte molecule is chemically unchanged during the polarisation process. An iridium-centered SABRE catalyst is typically employed to propagate polarisation, via J-coupling, between the *para*hydrogen derived hydrides and the spin−½ nuclei of the analyte molecule being polarised [17,18]. The most common pre-catalyst, [Ir(IMes)(COD)Cl] (**1**, IMes = 1, 3-bis (2, 4, 6-trimethylphenyl) imidazole-2-ylidene, COD = cyclooctadiene) has been shown to be an excellent catalyst for the hyperpolarisation of numerous clinically approved drugs such as nicotinamide [19,20,21,22], voriconazole [23,24], and niacin [25], among others [26,27].

One of the key goals of SABRE is its eventual application to in vivo imaging. Thus, any toxicity attributed to the SABRE catalyst, substrate, or solvent should be minimized or avoided. Substrate toxicity will have to be tested on an individual basis. But, as the SABRE process does not chemically change the substrate, previously approved drugs should be safe to use. Recent developments towards a biocompatible SABRE solvent system have also been reported through approaches including the use of water-soluble catalysts and pre-activating **1** to endow water solubility [22,28,29,30]. However, the highest enhancement levels have been reported when polarisation transfer occurs in a deuterated alcoholic solvent [31,32]. Therefore, a simple method suggested to generate a biologically compatible bolus is polarisation transfer in *d_6_*-ethanol, before dilution with D_2_O [33,34], with lower ethanol concentrations significantly decreasing mortality in mice [35]. Indeed, high ethanol concentrations have been utilized for the euthanasia of laboratory mice and rats [36,37,38], so work to determine a suitable dilution has been conducted by Duckett and co-workers [33,34].

The remaining, potentially toxic, component of the SABRE process is the iridium-containing polarisation transfer catalyst. Iridium salts have been reported to inhibit rat fibroblast cell proliferation in vitro in addition to inducing DNA damage [39]. The parenteral exposure limit for humans is 10 μg/day (10 ppm) [40]. Hypersensitivity to iridium salts has also been reported to be developed after long-term exposure in a factory setting [41]. Recently, Duckett et al. have sought to evaluate the toxicity of the SABRE process and determined that the main source of toxicity within a SABRE bolus is the presence of the catalyst [33]. The associated toxicity of the iridium complex means that techniques such as heterogeneous SABRE (HET-SABRE) catalysis [42,43,44], the use of biphasic mixtures to separate the catalyst and substrate [45], or making use of metal scavengers to remove the catalyst from solution are logical progressive steps towards clinical application [46].

Figure 1 shows a range of silica-supported scavengers, assessed as catalyst scavengers by Barskiy et al. [46]. Barskiy et al. began by making use of a biphasic mixture approach, using a mixed aqueous/organic phase to remove >99% of the activated iridium catalyst as the substrate (pyridine) favors the aqueous phase. There was, however, still a measurable quantity of iridium present in the aqueous phase, likely due to the reported water solubility of the catalyst once activated [22], which may prevent this approach from being used in isolation to produce a biocompatible bolus [47]. Barskiy et al. then assessed the metal scavenging efficacy of a range of nitrogen and sulfur-containing commercially available scavengers, which ranged in size widely from ~20–750 μm and loadings between 0.5 and ~2 mmol/g. Their work demonstrated that solid-supported scavengers could be effective at removing low concentrations of iridium over a 12-h period with 10 mg of the most effective scavenger removing ~95% of ~82 ppb iridium. The most effective scavenger was found to be Quadrasil MP which, alongside 2-mercaptoethyl ethyl sulfide silica, were added, along with unfunctionalized silica, to a mixed solvent system [48]. This combination resulted in catalyst capture within the *T*_1_ of a SABRE hyperpolarised substrate, in this case, metronidazole, resulting in an average of >98% catalyst removal, with Quadrasil MP still being the most effective scavenger reported. However, attempts to conduct SABRE following the addition of the scavengers were unsuccessful, with the authors noting the previously iridium-containing supernatant had become clear and the scavengers taking on a pale yellow color similar to the catalyst.

This work aims to establish the effectiveness of a range of scavengers for the rapid removal of [Ir(IMes)(pyridine)_3_(H)_2_] from the solution. Of particular interest was a systematic assessment of the impact of chain length, differing nitrogen-containing functional groups and the identity of the solid support employed. This work represents an important step towards the future in vivo application of SABRE, where high concentrations of cytotoxic iridium complexes may need to be rapidly removed in order to prepare a biologically compatible hyperpolarized bolus [33].

## 2. Results and Discussion

Previous work by Barskiy et al. reported on mainly functionalized silica supports to facilitate the removal of the iridium complex from the solution [44]. This work explores the use of different nitrogen-containing silica supports (Figure 2) to achieve the same aim. Substrates for testing were therefore chosen to examine if this approach was feasible for application in instances where secondary amine binding site(s) were available, heterocyclic rings were present and the identity of the solid support was varied from silica to polystyrene.

The range of scavengers utilized herein and presented in Figure 2 has the advantage of improved internal comparisons, in terms of particle size, when compared with the work of Barskiy et al. [46]. However, the difference in particle size and loadings limits the effectiveness of direct comparison between the two reports. When the support was not silica, the surface functionality was kept as similar as possible e.g., although supports **S_1_** and **S_2_** are functionalized silica and **S_6_** and **S_7_** are functionalized polystyrene, the functionalities are similar.

Characterization of the complex formed upon the surface of the scavenger would be challenging, therefore, solution-state analogs were utilized to investigate if these scavengers are able to deactivate the SABRE-active complex, from which scavenging potential can be inferred. As scavengers **S_1_**–**S_3_**, **S_6,_** and **S_7_** all contain a terminal amine and the binding of ammonia to the catalyst is well reported [49,50], ethylenediamine and diethylenetriamine were employed as direct analogs for ligand binding.

Following the addition of ethylenediamine or diethylenetriamine to a SABRE sample containing four equivalents of pyridine, relative to **1**, a significant reduction in polarisation, across all environments was observed −99.04% in the case of ethylenediamine and 99.55% in the case of diethylenetriamine. This decrease in polarisation demonstrates that the homogenous equivalent to supports **S_2_**, **S_3,_** and **S_7_** are effective at catalyst deactivation. These observations are in line with previous reports where bidentate ligands have been used for catalyst deactivation [51].

The displacement of pyridine from the activated species [Ir(IMes)(pyridine)_3_(H)_2_] was observed through monitoring the *T*_1_ values associated with pyridine, before and after the addition of ethylenediamine or diethylenetriamine as shown in Table 1.

Table 1 demonstrates that the relatively long *T*_1_ of pyridine in samples **A** and **B** is significantly reduced when in the presence of [Ir(IMes)(pyridine)_3_(H)_2_], shown in samples **C–E**, due to interactions with the extended spin system surrounding the iridium [52]. Notably, that although exchanges are brisk enough, with recorded pyridine exchange rates of 11.7 s^−1^ at 300 K [53], that the *T*_1_ of the pyridine free in solution and not currently bound to the catalyst is still drastically shortened compared to when the catalyst is not present.

The addition of ethylenediamine in sample **F** reverses much of the decrease in *T*_1_ caused by interaction with the iridium species. This observation is due to a majority (~90%) complex-forming (see Appendix A), in which ethylenediamine binds trans to the *para*hydrogen-derived hydrides. This orientation is analogous with those reported for bipyridine deactivation [51], and prevents pyridine (substrate) exchange. Notably, in this spatial orientation with ethylenediamine bound trans to hydride ligands, pyridine would still be bound trans to IMes. Ligands bound to this site have been reported not to exchange, so this is unlikely to impact the relaxation time of free pyridine [51].

When 2 equivalents of ethylenediamine were added relative to **1**, the deactivation process appears incomplete as demonstrated by the incomplete restoration of relaxation times to levels commensurate with isolated free pyridine and the continued presence of some SABRE signals. This is confirmed by the continued presence of minor hydride signals (see Appendix A) at −22.54 and −23.78 ppm at a ratio of ~5% each compared to the major species in solution. These chemical shift values are likely due to pyridine and methanol respectively bound *trans* to the *para*hydrogen derived hydrides. These peaks, however, are present at such low concentrations that full characterization of the complexes was not possible.

The addition of diethylenetriamine, shown in sample **G**, appears to deactivate the SABRE active species more effectively with an average relaxation time of 96.6% that of isolated pyridine; this value is commensurate with the 99.6% decrease in hyperpolarisation observed following diethylenetramine addition. This increase in deactivation efficacy may be due to a chelate effect with the tridentate diethylenetriamine forming a predominantly *fac*-isomer in solution, displacing pyridine from sites both trans to hydrides and the IMes ligand as shown in Appendix A. The formation of this isomer is supported through the lack of visible bound signals for pyridine in the ^1^H-NMR spectrum (see Appendix A).

The hydride observed at −22.68 is also similar to the reported signal attributed to [Ir(IMes)(NH_3_)_3_(H)_2_] in which the hydride resonance is present at −23.61 [49]. With the homogenous models demonstrating catalyst deactivation, **S_2_**, **S_3,_** and **S_7_** were assessed for their scavenging ability to progress SABRE towards in vivo application through removing the cytotoxic iridium species and aiding in the preparation of a biocompatible bolus by also acting as a catalyst scavenger [33].

Inductively coupled plasma (ICP) has previously been used for the determination of iridium removal from solution [46,48], therefore a method was designed for use with ICP-optical emission spectroscopy (OES). This method made use of the most sensitive Ir emission line at 224.268 nm with manual inspection of sample data ensuring there were no interferences [54]. The effect of organic components, i.e., pyridine, in the solutions to be tested, the required RF power, and plasma viewing mode were optimized (see Appendix A).

Scavengers **S_1_**, **S_2,_** and **S_3_** share a common structure, albeit with increasing potential binding sites: surprisingly **S_1_** was shown to be the most effective scavenging agent of these silica-based materials. Chelate effects would imply that **S_2_** and **S_3_** may form more stable complexes with the iridium SABRE catalyst. This interpretation is supported by a previous report by Mewis et al. wherein 2,2′-bipyridine and 1, 10-phenanthroline [51], which share a binding motif with scavenger **S_2_**, strongly bind to the SABRE active iridium species homogenously to effectively deactivate the catalyst [51]. However, it is possible that the steric bulk of the silica support prevents multiple binding sites from being favored. Scavenger **S_3_** was shown to be capable of scavenging the iridium catalyst from solution; however, the quantities removed were far lower than for **S_4_** (Figure 3), this comparison is in agreement with previous work of Barskiy et al. [46], who demonstrated ~81% removal of the iridium present within 12 h. Notably, **S_2_** appears to be a more effective scavenger than **S_3_** despite diethylenetriamine demonstrating more effective catalyst deactivation in a homogenous model. The reduced efficacy demonstrated here to sequester the catalyst may be due to the local steric bulk of the support preventing the formation of the *fac*-isomer characterized in Appendix A. Scavenger **S_1_** is demonstrated to be the most effective sequestering agent of **S_1_**–**S_3_**; the ability of this support to bind iridium has been previously reported in the context of synthesizing a HET-SABRE catalyst during which catalyst leaching into solution was not observed [44].

Although Figure 3 shows that only ~9.5% of the iridium was removed by S_3_ in this study within the first half-hour, it is worth noting that while the scavenger quantities are the same (10 mg of each), the initial iridium concentrations varied by orders of magnitude with this report’s initial concentration being ~3.25 ppm compared to the ~82 ppb in the solution tested by Barskiy et al. Therefore, despite the lower percentage value, ~294 ppb of iridium was removed by **S_3_** in this report within the first 32 min.

Scavenger **S_5_** is the most effective scavenger included in this test, continuing the theme of nitrogen-containing heterocyclic binding motifs that began with **S_4_**. This result represents a novel and potentially highly effective new scavenger that may be used in the preparation of an iridium-depleted bolus. Despite this, previous attempts to hyperpolarize 2-substituted picoline substrates have been unsuccessful [55]. This failure has been attributed to steric effects; however, 2-substituted pyridine compounds have been shown to bind to **1** through the binding of 2,2′-bipyridine and 1,10-phenanthroline, demonstrating that a sterically hindered pyridine may bind, apparently irreversibly [51]. In a separate report, Pravdivtsev notes that SABRE hyperpolarized signals for sites on bound 2,2′-bipyridine may be detected but evidence of this ligand exchanging is not presented [56].

Attempts to hyperpolarize 2-picoline as a homogenous analog of scavenger **5** were unsuccessful as reported in the literature [55]. However, when a co-ligand is utilized which may activate **1**, such as acetonitrile, then hyperpolarized signals corresponding to the ring of 2-picoline were evident with a maximum enhancement of 39.35 fold and 55.62 fold for the *para* proton when earths’ field and 65 G were used as the mixing field respectively (full enhancements in Appendix A). Duckett et al. have recently published similar results which utilize sulfoxide co-ligands to achieve polarization of 2-picoline [57].

Interestingly, scavengers **S_6_** and **S_7_** demonstrated essentially no scavenging effect, with Appendix A showing <5% iridium removal for both agents after a period of 12 days. As scavengers **S_1_** and **S_2_** are directly comparable with scavengers **S_6_** and **S_7_** in terms of binding modes, this significant difference in efficacy is likely due to the difference in support identity, polystyrene for scavengers **S_6_** and **S_7_** compared to silica for scavengers **S_1_**–**S_5_**. This observation is in line with the single polystyrene scavenger reported, which was the poorest of the scavengers so far assessed in the literature with steric effects of the support cited as the likely cause [42,46].

In this study, the most effective scavengers were **S_1_**, **S_4,_** and **S_5_**, which have all shown that 10 mg of scavenger may remove >1 ppm iridium over a long time period (12 days) and >60 ppb by t = 2 min. By t = 11 min these three scavengers had removed >115 ppb Ir from the solution.

Given the performance of scavengers **S_1_**, **S_4,_** and **S_5_**, it was decided to increase the concentrations of these scavengers above the 10 mg tested thus far. This increase in quantity used was to examine the effect of varying the number of scavengers on the speed of iridium scavenging.

Figure 4 presents an overview of data for variable scavenger masses of **S_1_**, **S_4,_** and **S_5_**. As expected, increasing the quantity of scavenger and exposure time both resulted in a marked decrease in iridium concentration remaining in solution for all three tested scavengers.

Scavenger **S_1_** performs well, with 100 mg able to sequester ~692 ppb of iridium in *t* = 2 min; however, the previously unreported scavenger **S_5,_** is able to remove ~939 ppb Ir in the same time period. The latter result represents a total removal of ~55% of the iridium present. This reduction in iridium presence may be compared with the results of Barskiy et al. whose best scavenger removes ~99.6% of the iridium present after a similar length of time [46]. It is not possible to compare directly the individual scavenger efficacy beyond overall percentage removal due to different particle sizes and loadings; this is demonstrated by comparing **S_B_** and **S_4_**, the former of which was shown to be most effective by others [46] whereas S_4_ is outperformed easily by **S_1_** and **S_5_** herein.

With a longer retention time, of t = 6 min a far higher quantity of iridium may be removed by scavenger **S_5_**, with ~1.4 ppm (~83%) removal. After t = 11 min this increases to 1.6 ppm (~94%) of the iridium present.

The quantity of scavenger **S_5_** required to achieve a level of iridium equal to the average value of the deionized water blanks (0.2126 ppb) run at the same time as these samples may be calculated through extrapolation of the data, as presented in Figure 5. Extrapolation utilized Equation S1 and assumes the trends observed over the measured quantities continue for higher scavenger masses.

Extrapolation of the data shown in Figure 5 shows that for t = 2 min about 1 g of scavenger **S_5_** would be required to remove ~1.7 ppm of iridium from solution under the conditions employed in this study. Recent reports have demonstrated that longer residence times are not necessarily a barrier to application towards an in vivo usage as biologically relevant drugs, such as the antibiotic metronidazole [58], may be hyperpolarized with a *T*_1_ of ~10 min allowing time for iridium removal [59,60]. This solid scavenging technique could be further integrated with complementary purification techniques, such as catalyst separated SABRE (CASH-SABRE), in order to formulate an efficient approach towards a clinical setting [45].

It is notable that the experimental method reported here is unlikely to be the most effective approach towards catalyst sequestering as limits on available SABRE pre-catalyst stock meant that samples for different time points were repeatedly vortexed and centrifuged. This centrifugation likely reduced the scavenging efficiency of the solids due to a reduction in sample homogeneity. It is envisioned that moving towards a flow-through system wherein centrifugation is not required would increase scavenging efficiency. The logical approach for preparing a metal-free bolus for injection would be packing a column and injecting the iridium-rich solution through a plug of the scavenger—this has been reported to have similar efficiency to “passive” scavenger exposure over a short retention time [46].

These data reported herein must be treated in the context of the SABRE experiment. As outlined in Table 1, entry G, the *T*_1_ of pyridine is restored in the presence of a deactivated catalyst. However, the *T*_1_ is still relatively short (25–33 s), especially if compared to other nuclei, such as ^15^N, for which *T*_1_ is minutes, as exemplified in the SABRE studies of metronidazole and nimrazole [26,60,61,62,63]. When the ^1^H *T*_1_ values of pyridine are considered alongside the shortest sampling time utilized for the ICP-OES measurements (2 min), it should be stated that 98% of the hyperpolarised signal would have been lost. This scenario also assumes that the *T*_1_ of pyridine is instantaneously elongated in the presence of the scavenger i.e., scavenging the catalyst has the same effect on *T*_1_ as catalyst deactivation. Evidently, such a scenario would not be applicable to a biological in vivo study due to the small amount of ^1^H hyperpolarised signal against a strong ^1^H-NMR background signal of the human body. The use of the methodology outlined herein is therefore not readily applicable to ^1^H but might be to ^15^N, given that ^15^N possesses long-lived states (relative to ^1^H) and biologically relevant analyte molecules e.g., metronidazole can be polarised by SABRE.

## 3. Materials and Methods

### 3.1. General

Scavengers **S_1_**–**S_7_**, deuterated solvents, and pyridine were all purchased from Sigma-Aldrich and used as received. **1** was synthesized according to the procedure of Sola et al. [64] ICP-OES analysis was performed using a Thermo Scientific iCAP 6000 series ICP-OES utilizing Qtegra software(Thermo Fisher Scientific, Waltham, MA, USA) and an iridium wavelength of 224.268 nm.

### 3.2. Apparatus Used for Generation of Parahydrogen

Hydrogen was generated using a Peak scientific PH200 hydrogen generator set to 3.5 bar. Where parahydrogen was required this was generated through the immersion of charcoal packed coil into liquid nitrogen to generate ~50% enriched *para*hydrogen.

### 3.3. SABRE Sample Preparation 

SABRE samples were prepared in J. Young’s tubes containing ~2mg **1**, the substrate, and co-ligands in 0.6 mL deuterated methanol. Samples were freeze-thaw degassed making use of a dry ice/acetone slush bath. SABRE measurements were undertaken by utilizing 3 bars of 50% *para*hydrogen and manual shaking for 10 s before taking a single scan at 1.4 T (Oxford Instruments Pulsar, Oxford Instruments, Abingdon, UK). Specifics of each sample are below, all equivalents relative to the actual value of 1 used (2.0–2.1mg).

For ethylenediamine/diethylenetriamine samples 4 equivalents of pyridine were initially added, freeze/thaw degassed, and SABRE observed before the addition of two equivalents of ethylenediamine or diethylenetriamine, freeze-thaw degassing and SABRE being attempted. *T*_1_ relaxation times were measured using a standard inversion recovery pulse sequence at 9.4 T.

For the 2-picoline sample, 4 equivalents of 2-picoline and 10 equivalents of acetonitrile were prepared, and SABRE was observed with shaking at either earth’s field or within an electromagnetic coil with current supplied to provide a local field of 65 G.

### 3.4. Preparation of ICP-OES Samples

A stock solution was made from 45.3 mg of **1** solvated in 2.4 mL d-methanol and 0.25 mL pyridine. This solution was sonicated (475H Langford sonomatic ultrasonic cleaner, Langford Ultrasonics, Bedworth, UK) for 1 min.

0.8 mL of this sample was transferred to three new Young’s tubes before being freeze-thaw degassed four times. Following degassing, 3 bar hydrogen was introduced into the tube and the tube was shaken for 1.5 min. Hydrogen was then refilled after 20 min, the tube shaken for a further minute and the tube left under hydrogen overnight.

Each of the solutions was then shaken for a further 1 min before being reduced to dryness under reduced pressure (with an additional ~0.3 mL d-methanol being used to wash out glassware and ensure complete transfer) before reconstitution with 8.5 mL deionized water *(*Merck Millipore gradient A10 water purification system, Merck Life Science, Watford, UK). 1 mL of each bolus was transferred to 8 sample tubes—one tube containing 10 mg of each scavenger and a tube without scavenger.

After a given sampling time, described in Appendix A, 0.1 mL of bolus was extracted and diluted with 9.9 mL of 3% HCl in deionized water to give a total sample volume of 10 mL for ICP-OES analysis.

This method was repeated exactly without the use of **1** to give three additional sets of samples, seven containing 10 mg of each scavenger and a tube containing no scavenger. 

In total, six samples per scavenger and six samples without scavenger were prepared, and within each set of six, three contained **1,** and three did not.

### 3.5. Preparation of ICP-OES Samples with Varying Catalyst Quantities

A stock solution was made from 15 mg of **1** solvated in 0.8 mL d-methanol and 82.5 µL pyridine. This solution was sonicated for 1 min. This solution was then transferred to three Young’s tubes prior to being degassed and hydrogen was introduced to the tube as described above.

The solution was then shaken for a further 1 min before being reduced to dryness under reduced pressure (with an additional ~0.3 mL d-methanol being used to wash out glassware and ensure complete transfer) before reconstitution with 24.5 mL deionized water. 1 mL of this bolus was added to 24 centrifuge tubes and 0, 10, 20, 30, 40, 50, 75, or 100 mg of scavengers S_1_, S_4,_ or S_5_ was added to each tube. This resulted in three tubes containing just the stock solution and three containing each scavenger at each concentration. Each tube was then vortexed for 10 s (VWR digital vortex mixer, VWR International, Lutterworth, UK) prior to being placed in a centrifuge for 2 min at 2500 rpm (VWR Microstar12 centrifuge, VWR International, Lutterworth, UK).

After a given sampling time, 0.1 mL of bolus was extracted and diluted with 9.9 mL of 3% HCl in deionized water to give a total sample volume of 10 mL for ICP-OES analysis.

This method was repeated without the use of **1** to give the same distribution of samples again without iridium present.

## 4. Conclusions

Removal of [Ir(IMes)(COD)Cl] from solution by scavenging may be a timely solution as recent work has demonstrated this iridium species is cytotoxic [33,34]. [Ir(IMes)(COD)Cl] is nonetheless still desirable for use as in most reports this pre-catalyst results in the highest enhancements [65,66]. Therefore, the potential for the supports to be utilized to remove this cytotoxic species from a bolus was examined.

The metal scavenging efficacy of the supports was examined and scavengers **S_1_**, **S_4,_** and **S_5_** were shown to be particularly effective. Although the work reported here represents a lower percentage removal of iridium from a SABRE bolus compared to the work of Barsiky et al. [46], the absolute concentration removed was significantly higher, making use of an initial iridium concentration closer to that which may be required in a clinical setting. The work presented herein also benefits from a significantly smaller range of particle sizes and loadings to facilitate more meaningful comparisons between scavengers tested.

**S_5_** was found to be the most promising scavenger and to the best of the author’s knowledge has not previously been employed for SABRE catalyst scavenging. Extrapolation of the scavenger data suggests that ~1 g of **S_5_** would render iridium below the level detected in blanks for the ICP-OES instrument utilized in this study, with an initial concentration of ~1.7 ppm and an exposure time of ~2 min. The quantity of scavenger required drops significantly with longer exposure times. It is envisaged that this novel scavenger could, in combination with other approaches such as CASH-SABRE [45], enable the rapid preparation of a biocompatible SABRE hyperpolarized bolus. Barskiy et al. demonstrated that a biphasic approach could remove 99–99.9% of iridium from a sample [46]. Therefore, utilization of their approach, combined with ~1g of **S_5_**, should be capable of reducing a sample commensurate with a typical SABRE sample (2 mg of **1** in 0.6 mL, ~1005 ppm) to background levels of iridium present within the hyperpolarized *T*_1_ of a suitable substrate [59,60].

## Figures and Tables

**Figure 1 molecules-27-00332-f001:**
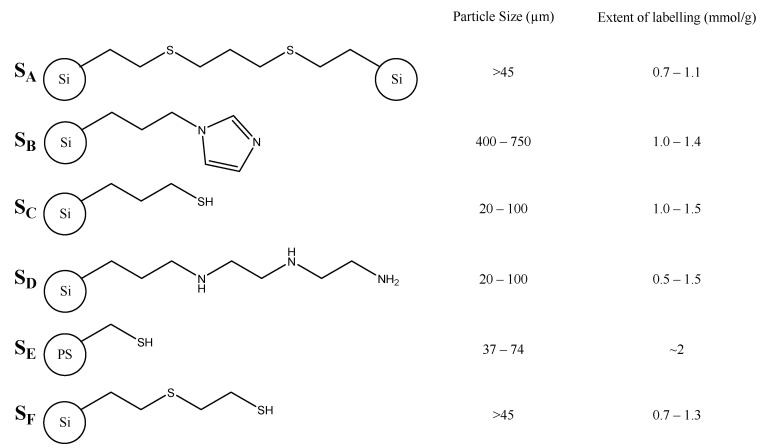
Scavengers tested by Barskiy et al. [46] **S_A_**–**S_F_** are silica mounted except **S_E_** which is polystyrene (PS) mounted.

**Figure 2 molecules-27-00332-f002:**
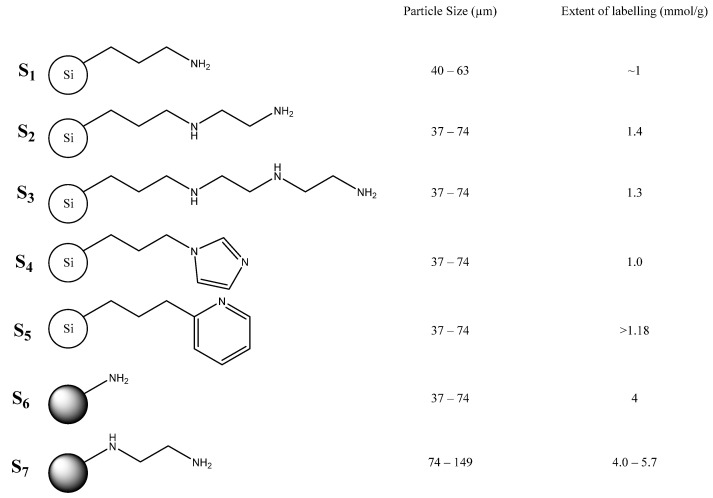
Scavengers tested within this work. Supports **S_1_**–**S_5_** are functionalized silica, **S_6_** and **S_7_** are functionalized polystyrene.

**Figure 3 molecules-27-00332-f003:**
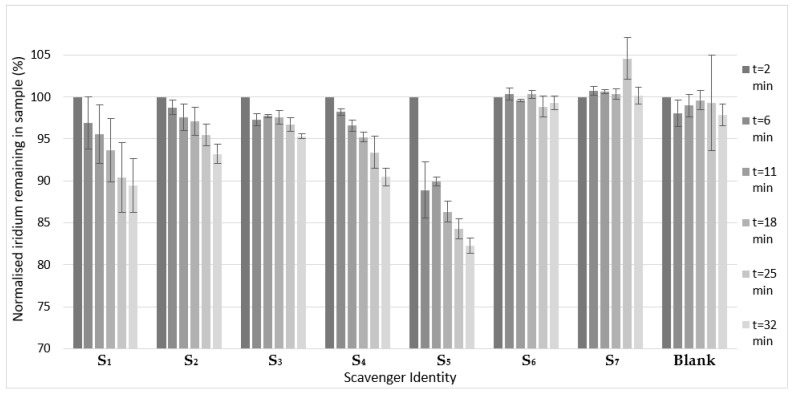
Iridium present in samples following the addition of 10 mg of scavengers **S_1_**–**S_7_** and with no scavenger added (**Blank**). Aliquots were taken over a range of time points with t = 2 min being the fastest experimentally feasible sampling time. Errors are one standard deviation between repeat samples (*N* = 3). For ease of comparison, the ordinate has been focused towards the top end of the scale.

**Figure 4 molecules-27-00332-f004:**
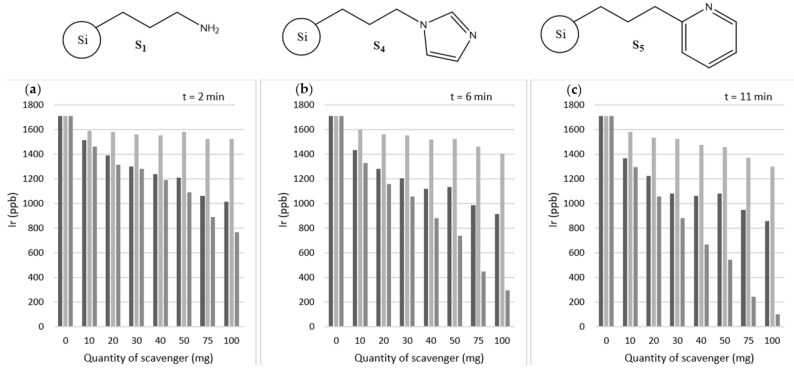
Amount of iridium present in samples following the addition of a variable amount of each scavenger (**S_1_** left column, **S_4_** middle column, **S_5_** right column) after (**a**) t = 2 min, (**b**) 6 min, and (**c**) 11 min exposure time (*N* = 1).

**Figure 5 molecules-27-00332-f005:**
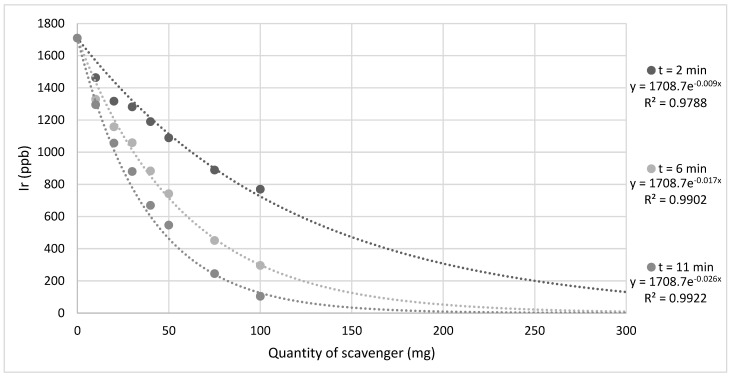
Amount of iridium present in samples following the addition of a variable amount of scavenger **S_5_** after *t* = 2 min, 6 min, and 11 min exposure time. The quantity of scavengers is extrapolated to demonstrate how much scavenger would be required for near-complete iridium removal.

**Table 1 molecules-27-00332-t001:** Pyridine *T_1_* values at 9.4 T demonstrating the effect of changing environment with and without **1**, ethylenediamine, and diethylenetriamine present. Sample ID from this table is utilized in the discussion.

Sample ID	[Ir(IMes)(COD)Cl] Present?	Atmosphere	Additional Ligand Added?	Pyridine Environment	Pyridine ^1^H Environment *T*_1_/s		
					*Ortho*	*Meta*	*Para*
**A**	No	Vacuum	No	Free	31.165	26.892	35.221
**B**	No	3 bar H_2_	No	Free	30.384	26.232	35.166
**C**	Yes	3 bar H_2_	No	Free	4.176	6.452	7.848
**D**	Yes	3 bar H_2_	No	Equatorially bound	4.044	5.897	9.494
**E**	Yes	3 bar H_2_	No	Axially bound	1.802	3.433	5.103
**F**	Yes	3 bar H_2_	Ethylenediamine (2 eq relative to **1**)	Free	15.094	25.316	18.871
**G**	Yes	3 bar H_2_	Diethylenetriamine (2 eq relative to **1**)	Free	30.030	25.422	33.079

## Data Availability

Data is contained within the article or Appendix A.

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
