# Peer review of "Rapid SABRE Catalyst Scavenging Using Functionalized Silicas"

_molecules, 2022, doi:10.3390/molecules27020332_

Round 1
Reviewer 1 Report
Robertson, Clarke and Mewis describe a systematic study to remove SABRE catalyst from HP solutions. SABRE technique has matured substantially over the past few years, and time is right for pilot in vivo studies. No such survival studies with SABRE technique has been demonstrated so far largely b/c robust preparation of HP solutions by SABRE has been challenging. A number of reports (that the co-authors thoroughly reviewed) have been described in literature with limited success. No protocol exists at the moment to prepare true biocompative solution of HP compounds by SABRE technique.
The Authors describe a systematic study that takes us one step closer to sich in vivo studies in the future (as a community). It is a very methodical and time-consuming study that is much needed for NMR/MRI hyperpolarization. As such, I rate the significance very high for this study, and this paper merits the publication in a solid peer-reviewed journal.
The paper is well written by experts in the field, and I recommend publication AS IS.
Author Response
Thanks to the reviewer for their comments on our work
Reviewer 2 Report
The submitted manuscript describes a systematic study of a catalyst scavenging method for iridium based SABRE hyperpolarization catalysts. The scavenging approach is based on a previously published manuscript from Barskiy et al. For this study, a set of scavengers is generated, focusing on different supports and chemical motives.
Generating an injectable solution is a crucial step towards medical applications of SABRE. Considering the low efficiency of existing heterogeneous catalysts, removing an efficient homogeneous catalyst after hyperpolarization seems to be the best way at the moment. The scavenging target therefore is the most studied and most efficient SABRE catalyst, which is iridium based and therefore highly toxic and not suitable for injection in medical applications. In this case though, fast and efficient removal of the catalyst are necessary, to yield maximum polarization and safety for injection. Unfortunately, the scavengers presented lack to provide both at the same time for most applications considering typical NMR relaxation times. Consequently, measuring hyperpolarized signals after extraction, the “ultimate goal” of such a project, is not part of the manuscript. Nonetheless, as the authors claim correctly, future improvements or combination with another removal method can lead to an extraction approach suitable for applications.
I support the idea of moving this scavenging method closer towards application and choosing such an analytical approach (by e.g. keeping the particle size and extent of labelling similar to focus on the involved chemistry). This deepens the understanding of the scavenging method and is an important step towards applications even though the system is not ready for direct applications yet.
That being said, I find some glaring issues with the current shape of the manuscript, especially its formal shape:
The manuscript is not very concise describing the work from Barskiy et al. as well as in the results section. Many parts are repetitive and lengthy and less pleasant to read compared to the other parts. The reiteration on previous work from Barskiy et al. spans an entire page and similar conclusions are drawn multiple times (e.g. that the initial iridium concentration is higher than in Barskiy et al. and therefore when comparing the studies, percentage removal is lower but amount removed higher, see line 250, 270 as well as line 385 of the conclusion)
The manuscript gives a total of 59 references mainly about SABRE hyperpolarization. Simon Duckett is certainly one of the (if not the) most important figure when it comes to SABRE hyperpolarization. Especially when considering the invention of SABRE and the major breakthroughs respect to the chemistry from his lab. Nonetheless, many labs around the world have been conducting numerous SABRE studies in the last decade. Thus, compared to other literature from the field, the number of citations from his group (21) seems a bit unbalanced. This may partly be because of the chemistry focused nature of the study and the previous position of Ryan Mewis, the last author of the manuscript, as a post-doc in Simon Ducketts team.
The T1 study in Table 1 is super interesting and strengthens your manuscript as a key source of information for your conclusions. Unfortunately, at first glance it is unclear how to read it. This could be improved. For example by giving each experiment a name or a numbers in an additional first column. In this way, it is also easy to refer to them in the text.
In many parts of the manuscript, the formal shape should be improved with respect to formatting, conciseness and visualization.
This also includes minor formatting issues such as not justified text parts (e.g. line 27-35 and 78-98), non italicized variables (e.g. in Figure 3 or line 281)and not well formatted references (e.g. ref 31 and 37) as well as style choices (line 329: “ten” instead of “10”; line 102: let the reader be the judge how important the step is?)
Especially the figures could use some rework. Figure 1 and 2 are crucial to be able to compare the efficiency of the scavengers in this work as well as the previous work from Barskiy et al. While reading for the first time, I found myself scrolling between Fig 1 and Fig 2 many times. I suggest you combine them into one figure yielding a stronger Figure 1. If you increase the element label sizes and at the same time the size of the scavenger drawings, they may even fit side by side.
Fig. 5 looks like a copy and paste excel plot that has not gotten much love. Why is there “t=0” equal to 2 minutes and “t=5” equal to 11 min? (Missing space in the caption “2mins” but it is clarified there) Maybe also draw the capture of iridium on Scavenger 5 on the top right of the figure. A graphical representation of the catalyst being captured somewhere in the manuscript would be a nice schematics.
SI:
Please also list and assign the other hydride at -23.7 ppm in Table S2.
Your assignment of the hydride signals in section 5 is very reasonable and most likely correct. Why don’t you draw the proposed structure and assign the NMR peaks to make it easier for the reader to visualize it?
Author Response
Reviewer 2
The submitted manuscript describes a systematic study of a catalyst scavenging method for iridium based SABRE hyperpolarization catalysts. The scavenging approach is based on a previously published manuscript from Barskiy et al. For this study, a set of scavengers is generated, focusing on different supports and chemical motives.
Generating an injectable solution is a crucial step towards medical applications of SABRE. Considering the low efficiency of existing heterogeneous catalysts, removing an efficient homogeneous catalyst after hyperpolarization seems to be the best way at the moment. The scavenging target therefore is the most studied and most efficient SABRE catalyst, which is iridium based and therefore highly toxic and not suitable for injection in medical applications. In this case though, fast and efficient removal of the catalyst are necessary, to yield maximum polarization and safety for injection. Unfortunately, the scavengers presented lack to provide both at the same time for most applications considering typical NMR relaxation times. Consequently, measuring hyperpolarized signals after extraction, the “ultimate goal” of such a project, is not part of the manuscript. Nonetheless, as the authors claim correctly, future improvements or combination with another removal method can lead to an extraction approach suitable for applications.
I support the idea of moving this scavenging method closer towards application and choosing such an analytical approach (by e.g. keeping the particle size and extent of labelling similar to focus on the involved chemistry). This deepens the understanding of the scavenging method and is an important step towards applications even though the system is not ready for direct applications yet.
Response: We thank the reviewer for their comments. No action taken
That being said, I find some glaring issues with the current shape of the manuscript, especially its formal shape:
The manuscript is not very concise describing the work from Barskiy et al. as well as in the results section. Many parts are repetitive and lengthy and less pleasant to read compared to the other parts. The reiteration on previous work from Barskiy et al. spans an entire page and similar conclusions are drawn multiple times (e.g. that the initial iridium concentration is higher than in Barskiy et al. and therefore when comparing the studies , percentage removal is lower but amount removed higher, see line 250, 270 as well as line 385 of the conclusion)
Response: The sections that detail the work covered by Barskiy has been significantly shortened. We do agree that this discussion is repetitious and so we have condensed the manuscript accordingly. Some examples of text changes are below:
Section deleted: notably more than the initial concentration tested by Barskiy et al. in their 12-hour as-sessments of scavenger efficacy.[45]
Section deleted: who demonstrated ~81% removal of the iridium present within 12 hours. This result ap-pears to be in good agreement with the data presented in figure 3. It is worth noting that a direct comparison cannot be made due to utilising different particle sizes and loadings.
Section deleted: in this work the sample was shaken for 10 seconds before 2 mins centrifugation com-pared to 10 seconds shaking, removal of an aliquot and 2 mins centrifugation in the work of Barskiy et al.[45] Although reporting a higher percentage removal, the initial Ir con-centration reported by Barskiy was 63.7 ppb, meaning the absolute iridium removed was 63.4 ppb, an order of magnitude below that demonstrated herein by scavenger S5.
Section changed from: It is not possible to compare directly the individual scavenger efficacy beyond overall percentage removal due to differing particle sizes and loadings; however, S4, the least ef-fective of the three scavengers tested in this study was employed by Barskiy et al. as SB (significantly higher particle size, similar loadings) and found to be one of the most effec-tive scavengers previously reported.[42] Comparing the efficacy of SB and S4, the latter is likely to have a higher surface area due to its smaller particle size, which may increase its scavenging efficacy, but it is still outperformed by S1 and S5 reported here.
To:
It is not possible to compare directly the individual scavenger efficacy beyond overall percentage removal due to differing particle sizes and loadings; this is demonstrated by comparing SB and S4, the former of which was shown to most effective by others [45] whereas S4 is out performed easily by S1 and S5 herein.
The manuscript gives a total of 59 references mainly about SABRE hyperpolarization. Simon Duckett is certainly one of the (if not the) most important figure when it comes to SABRE hyperpolarization. Especially when considering the invention of SABRE and the major breakthroughs respect to the chemistry from his lab. Nonetheless, many labs around the world have been conducting numerous SABRE studies in the last decade. Thus, compared to other literature from the field, the number of citations from his group (21) seems a bit unbalanced. This may partly be because of the chemistry focused nature of the study and the previous position of Ryan Mewis, the last author of the manuscript, as a post-doc in Simon Ducketts team.
Response: The reviewer does make a valid point about the distribution of references in that there are 21 references by Duckett. However, there are 11 citations for Chekmenev, 4 for Theis, 8 for Goodson, 5 for Barskiy and 6 from Koptyug (for example). The distribution of references is not influenced by the fact that 7 years ago, one of the authors was a post-doc in team of Simon Duckett. It is influenced by the relevance of those papers to the manuscript and the work contained within. The “distribution” of papers should reflect the papers that should be cited that aid the discussion. We note that the reviewer makes no indication of what references are missing – evidently key references have been cited (otherwise they should have been indicated) and thus the “imbalance” arises from the nature of the work discussed. However, as part of the review process we have added the following references (which are references 18, 26, 27, 40 and 41) to the manuscript to bring the total to 63, none of which are authored by Duckett
Barskiy, D.A.; Knecht, S.; Yurkovskaya, A.V.; Ivanov, K.L. SABRE: Chemical kinetics and spin dynamics of the formation of hyperpolarization. Progress in Nuclear Magnetic Resonance Spectroscopy 2019, 114-115, 33-70, doi:https://doi.org/10.1016/j.pnmrs.2019.05.005.
Salnikov, O.G.; Chukanov, N.V.; Svyatova, A.; Trofimov, I.A.; Kabir, M.S.H.; Gelovani, J.G.; Kovtunov, K.V.; Koptyug, I.V.; Chekmenev, E.Y. 15N NMR Hyperpolarization of Radiosensitizing Antibiotic Nimorazole by Reversible Parahydrogen Exchange in Microtesla Magnetic Fields. Angewandte Chemie International Edition 2021, 60, 2406-2413, doi:https://doi.org/10.1002/anie.202011698.
Kiryutin, A.S.; Yurkovskaya, A.V.; Petrov, P.A.; Ivanov, K.L. Simultaneous 15N polarization of several biocompatible substrates in ethanol–water mixtures by signal amplification by reversible exchange (SABRE) method. Magnetic Resonance in Chemistry 2021, 59, 1216-1224, doi:https://doi.org/10.1002/mrc.5184.
International Council for Harmonisation of Technical Requirements for Pharmaceuticals for Human Use, Guideline for Elemental Impurities Q3D(R1). Available online: https://www.ema.europa.eu/en/documents/scientific-guideline/international-conference-harmonisation-technical-requirements-registration-pharmaceuticals-human-use_en-32.pdf (accessed on November 2021).
Bergman, A.; Svedberg, U.; Nilsson, E. Contact urticaria with anaphylactic reactions caused by occupational exposure to iridium salt. Contact Derm. 1995, 32, 14-17, doi:doi:10.1111/j.1600-0536.1995.tb00833.x.
The T1 study in Table 1 is super interesting and strengthens your manuscript as a key source of information for your conclusions. Unfortunately, at first glance it is unclear how to read it. This could be improved. For example by giving each experiment a name or a numbers in an additional first column. In this way, it is also easy to refer to them in the text.
Response: A new column has been added to the table (Sample ID) and these are referred to in the text.
Reviewer 3 Report
This work addresses an important challenge in the field of parahydrogen-based hyperpolarization, namely the removal of the transition metal catalyst after performing the hyperpolarization procedure. At present, no general solution to this problem is available, and the demonstrated approaches are system-specific. The approach addressed in this work is based on metal complex scavenging with appropriately functionalized porous solids.
As this approach has been demonstrated before, the key question is whether a substantial advance is made in the study reported in this manuscript. In my opinion, this is not the case.
The authors state several times that
“Although the work reported here represents a lower percentage removal of iridium from a SABRE bolus compared to the work of Barskiy et al.[42], the absolute concentration removed was significantly higher, making use of an initial iridium concentration closer to that which may be required in a clinical setting.”
In other words, a lower percentage removal in combination with a much higher initial concentration means a much higher final concentration of Ir in the sample. However, it is this final concentration which matters for the potential in vivo applications, and not the quantity which was removed. For instance, it is seen in Figure 4 that 100 mg of the best scavenger removes a bit more than one half of the initial amount in 2 min, and even after 6 min only ~80% is removed, which is certainly not enough. Even longer scavenging times (e.g., 11 min) would be completely useless in practice (see below).
One of the key ideas of the study of Barskiy et al. was that catalyst scavenging was to be used as the secondary purification stage, after the initial Ir concentration is measurably reduced during the phase separation stage. This was the reason why they used a significantly lower initial Ir concentration in the scavenging experiments. In the end, the authors of this manuscript arrive to the same conclusion - that scavenging should be combined with other (e.g., phase separation) approaches.
In that case, why didn’t they test their scavengers with lower initial Ir concentrations, in particular to see if the scavengers they suggest are any better than those tested by Barskiy et al.? The fact that larger amounts of Ir are scavenged at higher initial concentrations does not automatically imply that higher amounts will be scavenged at lower starting values.
The next major issue which is not studied or discussed in this work is how the presence of solid scavengers would affect the relaxation of polarized molecules. While it is true that T1 times of 15N nuclei in some compounds can be rather long (up to 10 minutes), the relevant question is how long the T1 times will be during the prolonged and close contact with the porous solid support of the scavenging material. If T1 times are reduced to tens of seconds (which is possible, and even likely), the scavenging times of 6-11 minutes and more would be totally useless, and even a 2 min scavenging time may be too long and may reduce polarization dramatically.
In this respect, the comparison of 2 min scavenging in this work with 10 s shaking plus 2 min centrifugation in the study of Barskiy et al. is not appropriate and likely misleading - during centrifugation most of the substrate solution can be no longer in contact with the porous solid (Barskiy et al. used centrifugation to extract all the liquid from the pores, which is not necessary in applications), so that the actual contact time could have been more or less determined by the duration of sample shaking (~ 10 s).
Therefore, to make this work useful and interesting, it is imperative to test these scavengers (possibly, the best one) at lower initial Ir concentrations (and compare with the systems used by Barskiy) as well as demonstrate how much hyperpolarization survives after scavenging times that provide reasonable sample purity.
There is a number of less critical issues in the text of this manuscript.
It appears that the authors do not distinguish catalyst deactivation and catalyst scavenging, which is somewhat surprising. They use small-molecule analogues of their scavengers to explore their effect on the efficiency of Ir catalyst in the SABRE process. For instance:
“This decrease in polarization demonstrates that the homogenous equivalent to supports S2, S3 and S7 are effective at catalyst deactivation.”
This is also reflected in the manuscript title: “Rapid SABRE catalyst deactivation using functionalized silicas”
However, the main goal of these efforts is catalyst scavenging, which is certainly very different from deactivation.
And while deactivation implies binding of the Ir complex to the N-containing functionality of the scavenger, which interferes with the SABRE process, this binding is reversible and thus does not immediately imply efficient scavenging.
Some phrases need to be revised:
“catalyst capture within the lifetime of the substrate”
The lifetime of substrate is infinite for all practical purposes; it is a chemically stable molecule.
“An iridium centred spin-transfer catalyst”
It should be “iridium-centred”. Also, what is “spin-transfer catalyst”?
“with Quadrasil MP still being the most effective scavenger, compared to the theoretical maximum iridium concentration value.”
Scavenger cannot be compared to a concentration. Also, what is “theoretical maximum iridium concentration”?
“supports S1 and S2 are silica supported and S6 and S7 are polystyrene bound”
Supports are neither supported nor bound.
In Fig. 5, the scavenging times shown to the right of the graph are different from those specified in the caption, and t=0 makes no sense.
“When 2 equivalents of ethylenediamine were added relative to 1, the deactivation process appears incomplete … Addition of 2 equivalents of ethylenediamine results in near complete catalyst deactivation…”
These two back-to-back statements contradict each other.
Author Response
This work addresses an important challenge in the field of parahydrogen-based hyperpolarization, namely the removal of the transition metal catalyst after performing the hyperpolarization procedure. At present, no general solution to this problem is available, and the demonstrated approaches are system-specific. The approach addressed in this work is based on metal complex scavenging with appropriately functionalized porous solids.
As this approach has been demonstrated before, the key question is whether a substantial advance is made in the study reported in this manuscript. In my opinion, this is not the case.
The authors state several times that
“Although the work reported here represents a lower percentage removal of iridium from a SABRE bolus compared to the work of Barskiy et al.[42], the absolute concentration removed was significantly higher, making use of an initial iridium concentration closer to that which may be required in a clinical setting.”
In other words, a lower percentage removal in combination with a much higher initial concentration means a much higher final concentration of Ir in the sample. However, it is this final concentration which matters for the potential in vivo applications, and not the quantity which was removed. For instance, it is seen in Figure 4 that 100 mg of the best scavenger removes a bit more than one half of the initial amount in 2 min, and even after 6 min only ~80% is removed, which is certainly not enough. Even longer scavenging times (e.g., 11 min) would be completely useless in practice (see below).
Response: We think the reviewer has missed the point of the work. For an in vivo bolus, a large amount of substrate would be required and this would require a large amount of catalyst to effect hyperpolarisation. Therefore, the initial amount of catalyst is important, as is the final concentration. The toxicity of iridium has been assessed to be 10 μg/day (added to manuscript lines 68 and 69), and thus preparing a sample below this cytotoxic threshold is important in terms of moving forwards. Of the scavengers tested, S5, achieved a threshold lower than 800 ppb (0.8 ppm), an order of magnitude lower than the toxicity threshold. Thus the amount removed is entirely relevant for use and not “completely useless in practice”.
One of the key ideas of the study of Barskiy et al. was that catalyst scavenging was to be used as the secondary purification stage, after the initial Ir concentration is measurably reduced during the phase separation stage. This was the reason why they used a significantly lower initial Ir concentration in the scavenging experiments. In the end, the authors of this manuscript arrive to the same conclusion - that scavenging should be combined with other (e.g., phase separation) approaches.
In that case, why didn’t they test their scavengers with lower initial Ir concentrations, in particular to see if the scavengers they suggest are any better than those tested by Barskiy et al.? The fact that larger amounts of Ir are scavenged at higher initial concentrations does not automatically imply that higher amounts will be scavenged at lower starting values.
Response: The initial concentration in the work of Barskiy was originally relatively high – hence why the scavenging process occurred as a second purification stage as indicated by the reviewer in the paragraph above. This means the initial concentration was higher and therefore the second purification was purely concerned with the residual element. This work scavenges the SABRE catalyst at a level that is commensurate with the original SABRE catalysis i.e. scavenging is the first stage of catalyst removal. Therefore, the use of smaller concentrations is of little relevance here because the two approaches are not the same; the use of a higher concentration reflects that that might be used to prepare a medically relevant bolus.
The next major issue which is not studied or discussed in this work is how the presence of solid scavengers would affect the relaxation of polarized molecules. While it is true that T1 times of 15N nuclei in some compounds can be rather long (up to 10 minutes), the relevant question is how long the T1 times will be during the prolonged and close contact with the porous solid support of the scavenging material. If T1 times are reduced to tens of seconds (which is possible, and even likely), the scavenging times of 6-11 minutes and more would be totally useless, and even a 2 min scavenging time may be too long and may reduce polarization dramatically.
Response: The reviewer makes a very relevant point. Sadly, in the very short time afforded to make corrections to the manuscript, such studies cannot be conducted. However, and perhaps more importantly, how would one measure a T1 of such a solution when the porous solid support will naturally gravitate to the bottom of the NMR tube, and therefore be outside the “window” of analysis, whereas “free” molecules will be, with their T1s most likely completely unaffected? This is not a measurement that can be simply made using solution-based NMR whereby meaningful data is obtained. Spinning of the sample could be used, but this would result in a heterogeneous solution leading to very poor lineshape and unreliable data.
In this respect, the comparison of 2 min scavenging in this work with 10 s shaking plus 2 min centrifugation in the study of Barskiy et al. is not appropriate and likely misleading - during centrifugation most of the substrate solution can be no longer in contact with the porous solid (Barskiy et al. used centrifugation to extract all the liquid from the pores, which is not necessary in applications), so that the actual contact time could have been more or less determined by the duration of sample shaking (~ 10 s).
Response: The two mins scavenging time (actual time is 2 mins 10 s as identified in table S3) is the quickest time whereby a sample can be constituted for measurement. This 2 mins 10s consists of 10s vortexing and 2 mins of centrifugation. Longer time points were used to investigate the effect over time, as identified in the manuscript. If there was no difference in this approach, one would expect the same values to be obtained irrespective of the time index. As showcased many times, this isn’t the case. Figure S7 showcases the methodology employed.
Therefore, to make this work useful and interesting, it is imperative to test these scavengers (possibly, the best one) at lower initial Ir concentrations (and compare with the systems used by Barskiy) as well as demonstrate how much hyperpolarization survives after scavenging times that provide reasonable sample purity.
Response: As stated above, this request does not fit with the objectives of the manuscript and is therefore a redundant request. The focus on lower initial concentrations is not relevant to the study presented
There is a number of less critical issues in the text of this manuscript.
It appears that the authors do not distinguish catalyst deactivation and catalyst scavenging, which is somewhat surprising. They use small-molecule analogues of their scavengers to explore their effect on the efficiency of Ir catalyst in the SABRE process. For instance:
“This decrease in polarization demonstrates that the homogenous equivalent to supports S2, S3 and S7 are effective at catalyst deactivation.”
This is also reflected in the manuscript title: “Rapid SABRE catalyst deactivation using functionalized silicas”
However, the main goal of these efforts is catalyst scavenging, which is certainly very different from deactivation.
And while deactivation implies binding of the Ir complex to the N-containing functionality of the scavenger, which interferes with the SABRE process, this binding is reversible and thus does not immediately imply efficient scavenging.
Response: We understand the point that the reviewer has made. We demonstrate catalyst deactivation using ethylenediamine and diethylenetrimaine. We highlight that for the former some SABRE signals persist. The scavenging process is different, and so we have made necessary changes:
Title changed to “Rapid SABRE scavenging using functionalized silicas”
Sentence highlighted above by reviewer changed to “With the homogenous models demonstrating catalyst deactivation, S2, S3 and S7 were assessed for their scavenging ability to progress SABRE towards an in vivo application…..”
Some phrases need to be revised:
“catalyst capture within the lifetime of the substrate”
The lifetime of substrate is infinite for all practical purposes; it is a chemically stable molecule.
Response: Changed to “….within the T1 of a SABRE hyperpolarised substrate, in this case metronidazole”
“An iridium centred spin-transfer catalyst”
It should be “iridium-centred”. Also, what is “spin-transfer catalyst”?
Response: Changed to “iridium-centred SABRE catalyst”
“with Quadrasil MP still being the most effective scavenger, compared to the theoretical maximum iridium concentration value.”
Scavenger cannot be compared to a concentration. Also, what is “theoretical maximum iridium concentration”?
Response: Changed to “, with Quadrasil MP still being the most effective scavenger reported”
“supports S1 and S2 are silica supported and S6 and S7 are polystyrene bound”
Supports are neither supported nor bound.
Response: Sentence changed to “… supports S1 and S2 are functionalized silicas and S6 and S7 are functionalized polystyrenes….”
In Fig. 5, the scavenging times shown to the right of the graph are different from those specified in the caption, and t=0 makes no sense.
Response: Figure updated. Values of t now equal 2, 6 and 11 min.
“When 2 equivalents of ethylenediamine were added relative to 1, the deactivation process appears incomplete … Addition of 2 equivalents of ethylenediamine results in near complete catalyst deactivation…”
These two back-to-back statements contradict each other.
Response: The second sentence beginning “addition of 2 equivalents of ethylenediamine….” has been removed as it is contradictory, as the reviewer has indicated. The section now reads as “When 2 equivalents of ethylenediamine were added relative to 1, the deactivation process appears incomplete as demonstrated by the incomplete restoration of relaxation times to levels commensurate with isolated free pyridine and the continued presence of some SABRE signals. This is confirmed by the continued presence of minor hydride signals (see figure S2) at -22.54 and -23.78 ppm at a ratio of ~5% each compared to the major species in solution.”
Round 2
Reviewer 3 Report
I’ve considered carefully the authors’ responses to my comments and the changes in the manuscript text. I can agree to some (but not all) of their reasoning,.
One of the authors’ responses states:
“For an in vivo bolus, a large amount of substrate would be required and this would require a large amount of catalyst to effect hyperpolarisation. Therefore, the initial amount of catalyst is important, as is the final concentration. The toxicity of iridium has been assessed to be 10 μg/day (added to manuscript lines 68 and 69), and thus preparing a sample below this cytotoxic threshold is important in terms of moving forwards. Of the scavengers tested, S5, achieved a threshold lower than 800 ppb (0.8 ppm), an order of magnitude lower than the toxicity threshold. Thus the amount removed is entirely relevant for use and not “completely useless in practice””.
Ok, I admit that “completely useless” may have been a poor choice of wording. What was meant there is that the final concentration and the bolus volume are the key factors for a safe injection. The rest is a matter of the process scale-up – larger amounts of solution would require larger amounts of catalyst and larger amounts of a scavenger.
However, there are several problems with the statements cited above. First of all, the document that the authors now cite as ref. [40] also states that “There is limited toxicological data for the Platinum-Group Elements (PGE) other than platinum, and, to a lesser extent, palladium.” Therefore, the 10 μg/day safety limit is yet to be verified. Also, it is not clear why the authors compare the concentration (0.8 ppm) with the dose (10 μg/day).
Overall, however, this discussion of amounts, concentrations and scale-up, once properly refined, could be acceptable in the end.
Nevertheless, my initial assessment of the work is not changed much by the authors’ replies and changes they implemented in the manuscript.
To the following comment:
“Therefore, to make this work useful and interesting, it is imperative to test these scavengers (possibly, the best one) at lower initial Ir concentrations (and compare with the systems used by Barskiy) as well as demonstrate how much hyperpolarization survives after scavenging times that provide reasonable sample purity.”
the authors respond:
“Response: As stated above, this request does not fit with the objectives of the manuscript and is therefore a redundant request.”
Putting aside the issue of concentrations (as mentioned above), this essentially means that the authors consider the question of how much hyperpolarization could survive the scavenging process as redundant and being beyond the scope of this work. I find this impossible to understand and/or accept.
So what is then the key objective of this work? To demonstrate that the Ir catalyst can be efficiently scavenged given an unlimited amount of time and scavenger? Quite likely, there would not be much interest and novelty in this.
However, if the authors intend to put their study in the context of SABRE, it is an entirely different story, as now the allowed scavenging time and the amount of scavenger are bound to be severely limited.
In fact, this work and its results are, without doubt, positioned as an important development in the context of SABRE research; for instance:
“This work represents an important step towards the future in vivo application of SABRE, where high concentrations of cytotoxic iridium complexes may need to be rapidly removed in order to prepare a biologically compatible hyperpolarized bolus.”
and also in one of the replies from the authors:
“Therefore, the use of smaller concentrations is of little relevance here because the two approaches are not the same; the use of a higher concentration reflects that that might be used to prepare a medically relevant bolus.”
However, this work shows only that a biologically compatible sample can be prepared, but provides no information whatsoever whether this sample has any chance to remain hyperpolarized. Lost polarization would make the bolus biologically compatible but completely useless (and in this case, this is not an exaggeration).
To reiterate, once the authors put their work in the context of SABRE research (which is likely the only meaningful context anyway), the primary question that inevitably emerges is: what is the relevant timescale after which all this becomes impractical? This is certainly not an easy question to answer, and the result will depend on many factors, with the amount of scavenger and fluid volume used being among the most important ones. Without establishing the relevant timescale and demonstrating that scavenging can be performed rapidly on that timescale, the results of this work are largely incomplete and cannot be accepted for publication in my opinion. The argument that there was not enough time to look into this issue of fundamental importance is a rather poor one. As for how such measurements could be performed, the most straightforward way would be to produce a hyperpolarized molecule and then apply the scavenging procedure. As the authors are equipped to perform SABRE experiments, they need to demonstrate a sincere effort toward achieving thus rather than undermine the importance of this issue which is of utmost importance in the context of SABRE research.
Author Response
One of the authors’ responses states:
“For an in vivo bolus, a large amount of substrate would be required and this would require a large amount of catalyst to effect hyperpolarisation. Therefore, the initial amount of catalyst is important, as is the final concentration. The toxicity of iridium has been assessed to be 10 μg/day (added to manuscript lines 68 and 69), and thus preparing a sample below this cytotoxic threshold is important in terms of moving forwards. Of the scavengers tested, S5, achieved a threshold lower than 800 ppb (0.8 ppm), an order of magnitude lower than the toxicity threshold. Thus the amount removed is entirely relevant for use and not “completely useless in practice””.
Ok, I admit that “completely useless” may have been a poor choice of wording. What was meant there is that the final concentration and the bolus volume are the key factors for a safe injection. The rest is a matter of the process scale-up – larger amounts of solution would require larger amounts of catalyst and larger amounts of a scavenger.
However, there are several problems with the statements cited above. First of all, the document that the authors now cite as ref. [40] also states that “There is limited toxicological data for the Platinum-Group Elements (PGE) other than platinum, and, to a lesser extent, palladium.” Therefore, the 10 μg/day safety limit is yet to be verified. Also, it is not clear why the authors compare the concentration (0.8 ppm) with the dose (10 μg/day).
Overall, however, this discussion of amounts, concentrations and scale-up, once properly refined, could be acceptable in the end.
Nevertheless, my initial assessment of the work is not changed much by the authors’ replies and changes they implemented in the manuscript.
RESPONSE: The does of 10 ug per day is equivalent to 10 ppm as stated on line 69. Thus achieving a level of 0.8 ppm in the sample is below the threshold. As the reviewer points out, there is evidence of toxicological data – we cannot comment on the verifiable nature of this data as the manuscript presents no results to this end.
To the following comment:
“Therefore, to make this work useful and interesting, it is imperative to test these scavengers (possibly, the best one) at lower initial Ir concentrations (and compare with the systems used by Barskiy) as well as demonstrate how much hyperpolarization survives after scavenging times that provide reasonable sample purity.”
the authors respond:
“Response: As stated above, this request does not fit with the objectives of the manuscript and is therefore a redundant request.”
Putting aside the issue of concentrations (as mentioned above), this essentially means that the authors consider the question of how much hyperpolarization could survive the scavenging process as redundant and being beyond the scope of this work. I find this impossible to understand and/or accept.
So what is then the key objective of this work? To demonstrate that the Ir catalyst can be efficiently scavenged given an unlimited amount of time and scavenger? Quite likely, there would not be much interest and novelty in this.
However, if the authors intend to put their study in the context of SABRE, it is an entirely different story, as now the allowed scavenging time and the amount of scavenger are bound to be severely limited.
In fact, this work and its results are, without doubt, positioned as an important development in the context of SABRE research; for instance:
“This work represents an important step towards the future in vivo application of SABRE, where high concentrations of cytotoxic iridium complexes may need to be rapidly removed in order to prepare a biologically compatible hyperpolarized bolus.”
and also in one of the replies from the authors:
“Therefore, the use of smaller concentrations is of little relevance here because the two approaches are not the same; the use of a higher concentration reflects that that might be used to prepare a medically relevant bolus.”
However, this work shows only that a biologically compatible sample can be prepared, but provides no information whatsoever whether this sample has any chance to remain hyperpolarized. Lost polarization would make the bolus biologically compatible but completely useless (and in this case, this is not an exaggeration).
To reiterate, once the authors put their work in the context of SABRE research (which is likely the only meaningful context anyway), the primary question that inevitably emerges is: what is the relevant timescale after which all this becomes impractical? This is certainly not an easy question to answer, and the result will depend on many factors, with the amount of scavenger and fluid volume used being among the most important ones. Without establishing the relevant timescale and demonstrating that scavenging can be performed rapidly on that timescale, the results of this work are largely incomplete and cannot be accepted for publication in my opinion. The argument that there was not enough time to look into this issue of fundamental importance is a rather poor one. As for how such measurements could be performed, the most straightforward way would be to produce a hyperpolarized molecule and then apply the scavenging procedure. As the authors are equipped to perform SABRE experiments, they need to demonstrate a sincere effort toward achieving thus rather than undermine the importance of this issue which is of utmost importance in the context of SABRE research.
RESPONSE: We recognise that the reviewer has a series of valid points that relate to fully translating this approach. We have tried to position our work in the context of others and to indicate that this approach is not the complete answer to removing the SABRE catalyst but part of a wider solution that will involve other methodologies. Hence why we have indicated that it is an important step (and not the complete answer) in preparing a medically relevant bolus. To do so would need nuclei with longer T1s i.e. not proton to be effective. To this end, the following paragraph has been added to the end of the results and discussion to highlight the drawbacks of the approach detailed and to highlight how the work could be used by others (we lack the capacity to do 15N on our benchtop, for example):
These data reported herein must be treated in the context of the SABRE experiment. As outlined in Table 1, entry G, the T1 of pyridine is restored in the presence of a deactivated catalyst. However, the T1 is still relatively short (25-33 s), especially if compared to other nuclei, such as 15N, for which T1 is minutes, as exemplified in the SABRE studies of metronidazole and nimrazole.[26,60-63] When the 1H T1 values of pyridine are considered alongside the shortest sampling time utilised for the ICP-OES measurements (2 min), it should be stated that 98% of the hyperpolarised signal would have been lost. This scenario also assumes that the T1 of pyridine is instantaneously elongated in the presence of the scavenger i.e. scavenging the catalyst has the same effect on T1 as catalyst deactivation. Evidently, such a scenario would not be applicable to a biological in vivo study due to the small amount of 1H hyperpolarised signal against a strong 1H NMR background signal of the human body. The use of the methodology outlined herein is therefore not readily applicable to 1H but might be to 15N, given that 15N possesses long lived states (relative to 1H) and biologically relevant analyte molecules e.g. metronidazole can be polarised by SABRE.